# HyperLogic: Enhancing Diversity and Accuracy in Rule Learning with HyperNets

**Yang Yang[1], Wendi Ren[1], Shuang Li[1]***
[1]School of Data Science, The Chinese University of Hong Kong (Shenzhen)
{yangyang8, wendiren}@link.cuhk.edu.cn, lishuang@cuhk.edu.cn

## Abstract

Exploring the integration of if-then logic rules within neural network architectures presents an intriguing area. This integration seamlessly transforms the rule learning task into neural network training using backpropagation and stochastic gradient descent. From a well-trained sparse and shallow neural network, one can interpret each layer and neuron through the language of logic rules, and a global explanatory rule set can be directly extracted. However, ensuring interpretability may impose constraints on the flexibility, depth, and width of neural networks. In this paper, we propose HyperLogic: a flexible approach leveraging hypernetworks to generate weights of the main network. HyperLogic can be combined with existing differentiable rule learning methods to generate diverse rule sets, each capable of capturing heterogeneous patterns in data. This provides a simple yet effective method to increase model flexibility and preserve interpretability. We theoretically analyze the benefits of the HyperLogic by examining the approximation error and generalization capabilities under two types of regularization terms: sparsity and diversity regularization. Experiments on real data demonstrate that our method can learn more diverse, accurate, and concise rules. Our code is publicly available at https://github.com/YangYang-624/HyperLogic.

## 1 Introduction

Despite the significant impact of deep learning on society, its lack of interpretability limits its use in critical areas that demand high transparency. For instance, in high-risk domains such as healthcare, finance, and law, the decision-making process of models needs to be fully open and explainable to users and relevant regulatory authorities to gain necessary trust and legitimacy [1, 2, 3]. Compared to the "black box" models, which may perform well but are uninterpretable, people prefer intrinsically interpretable model for decision support [4, 5], such as a set of concise IF-THEN rules.

Traditional rule learning methods, which are based on statistical or heuristic approaches, have been extensively explored but often struggle to simultaneously achieve two key objectives: 1) simplicity and accuracy in rules, and 2) noise tolerance and scalability in data handling [6, 7, 8, 9, 10]. These methods typically rely on search-based techniques, which can be limited when dealing with complex and noisy data. In contrast, deep learning, which focuses on representation learning through embedding, is robust to noise and effectively manages large datasets. This has led to the exploration of differentiable rule learning [11], which leverages the high performance of deep learning while ensuring interpretability through explicit rule formulation

Differentiable rule learning primarily includes two types of methods. The first approach outputs rules directly through the network, leveraging powerful and diverse neural models to enhance handling of complex data patterns [12, 13, 14]. The second approach is to extract rules from network weights [15, 16, 17], which promotes rapid training and efficient data management while enabling the use of predefined structures to integrate expert priors [18].

---

*Corresponding author

38th Conference on Neural Information Processing Systems (NeurIPS 2024).

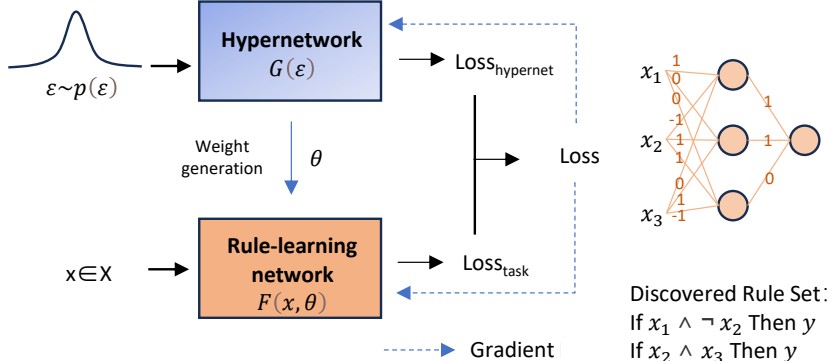

Figure 1: The framework of HyperLogic: Hypernetwork generates $\theta$ for the main network, which is a rule-learning network. An example of a main rule-learning network is shown on the right, from which rules can be extracted based on the learned network weights.

We focus on the second approach, aiming to extract rules like "IF $a \wedge b \wedge \neg c$ THEN $y$ is True" from network weights, where $a, b, c$ are predefined predicates. Although integrating if-then logic rules within neural network architectures offers clear advantages, their performance is often hindered by the restrictive network structures required for interpretability. These structures tend to be overly simplistic and not scalable, limiting their ability to capture complex data patterns and making them prone to only capturing partial or local patterns in the data. This raises a critical question: How can we modify these models to better harness the full potential of neural networks without compromising their interpretability?

In this work, we introduce **HyperLogic**, a novel framework that integrates hypernetworks with a main rule-learning network (as illustrated in Fig. 1, where an example of a main rule-learning network is shown on the right). Hypernetworks are a type of neural network that generate weights for a main rule-learning network. In our framework, the hypernetwork takes random samples from a high-dimensional Gaussian distribution as input and outputs weights for different parts of the main network. These weights can be utilized in two ways: either directly as the weights for the main network throughout the training process (meaning the main network itself has no parameters), or by retaining trainable weights in the main network and combining them with the hypernetwork-generated weights through weighted fusion. In both scenarios, this approach yields an interpretable set of rules derived from the comprehensive weights.

HyperLogic can be seen as a mixture of expert model with an infinite number of experts, providing a simple yet effective way to enhance model flexibility and adaptability. We analyze the benefits of HyperLogic by examining its approximation error and generalization capabilities under two types of regularization: sparsity and diversity regularization. Our findings demonstrate that HyperLogic acts as a universal approximation estimator in the Barron space [19] and proves its generalization ability across various regularization methods. These theoretical benefits are further supported by our empirical experiments.

Additionally, hypernetworks enable the generation of multiple candidate rule sets in a single training session without significantly increasing computational overhead. This is in stark contrast to traditional methods, which typically learn only one set of rules at a time. Consequently, HyperLogic greatly enhances the efficiency and flexibility of the rule-learning process.

We summarize our contributions as follows:

1. We introduce HyperLogic, a pioneering framework that integrates hypernetworks into differentiable rule learning, significantly enhancing the rule-learning landscape.

2. We theoretically justify the performance of HyperLogic and provide insights into its effectiveness. Specifically, we examine the approximation error and generalization capabilities under two types of regularization: sparsity and diversity.

3. We validate HyperLogic through extensive experiments on multiple datasets, demonstrating that it enables the learning of multiple diverse rule sets and yields more concise and accurate rules.

## 2   Related Work

**Learning Rules from Network Weights**   Many methods for extracting rules from model weights are not based on neural networks; instead, they primarily focus on interpreting the weights of smaller,

simpler models to discover rules [20, 7]. Recently, extensive methods have emerged that attempt to extract rules using neural network models. These approaches often involve data preprocessing techniques to prepare the data before training [21] or utilize post-hoc methods for rule extraction after model training [22, 23, 24], which can often lead to a loss of accuracy.

A more promising alternative allows for the direct interpretation of neural network weights as precise rules by integrating rule extraction directly during training. This leads to a more seamless and effective learning process. For example, methods such as [25, 26, 27] utilize low-dimensional embeddings to represent atomic conditions and rules, treating rule learning as a differentiable discrete combinatorial optimization problem encoded by a feedforward neural network. Specifically, FinRule [27] explores higher-order interactions between atomic conditions, which aligns somewhat with our approach of employing hypernetworks. However, it lacks a detailed analysis of the specific roles of these higher-order interactions and cannot learn a rich set of candidate rules as our method does. Additionally, these approaches often impose constraints on rule templates, such as fixing the rule length and the number of rules.

Another common class of methods employs modified simple feedforward neural networks, such as DR-net [17] and others [28, 29, 30]. These methods simulate logical operations by altering activation functions and implementing mechanisms that ensure differentiability and support backpropagation. While they overcome the limitations of rigid rule templates, their model structures tend to be relatively simple. Furthermore, the training process often involves freezing the weights of one component while adjusting another, which restricts the model's ability to fully optimize performance.

Our HyperLogic is a unified framework that can be integrated with various neural network weight generators, enabling easy adaptation to different main rule learning networks. Currently, our main network draws inspiration from DR-net [17], but it can be replaced with other architectures based on task requirements. Unlike recent Bayesian approaches, which often require multiple hyperparameters to model uncertainty in network weights, HyperLogic simplifies the process by needing fewer hyperparameters while maintaining scalability. This allows for efficient training and adaptability across various tasks without the added complexity of managing uncertainty. Additionally, our approach can generate multiple sets of rule sets in a single training session, a capability not available in other methods. For further comparisons, including classic rule-based algorithms, please refer to Appendix A.

**Hypernetwork**  Hypernetworks, or hypernets, are neural networks that generate weights for main networks [31]. Benefiting from over-parameterization and strategic design, these networks enhance training flexibility and adaptability, improve information sharing, and accelerate training processes. While widely used in many areas like continual learning [32, 33], transfer learning [34], uncertainty quantification [35, 36], natural language processing [37] and computer vision [38], their potential in rule learning remains largely untapped. Our HyperLogic framework can bridge this gap. Specifically, our hypernetwork takes inspirations from HyperGAN [36] and similarly we add a loss term to encourage the diversity of the generated network weights.

Unlike typical applications in other domains that focus on enhancing parameter efficiency and training speed, our approach addresses the unique challenges of adding model flexibility while preserving interpretability. Our main goal is to expand the parameter space to produce more diverse, concise and accurate rule sets.

## 3   HyperLogic

HyperLogic is a versatile framework designed to enhance various existing neural rule-learning methods. The concept of hypernetworks can be applied to different types of main networks, provided they focus on learning a set of interpretable weights in a differentiable manner. Importantly, HyperLogic imposes no additional restrictions on data formats or rule languages; it simply builds upon the capabilities of the chosen main network.

### 3.1   Main Network: Rule-Learning Network

Our main network is currently based on DR-Net [17], which features a simple architecture and is designed for binary classification tasks. We chose this architecture for its simplicity, which aids in our proofs and explanations. However, we recognize its limitations, so we can also explore other more flexible neural approaches as the primary network to overcome restrictions related to data formats and rule languages. This is further discussed in Appendix B.

The main network is a two-layer neural network as shown in Fig. 1 (right part). Due to its interpretability, one can directly extract the if-then rules from the trained neural networks. The input layer of the main network is fed with a $D$-dimensional binary data $x = [x_d]$, each element $x_d \in \{1, -1\}$. Here $x_d$ indicates the grounded predicate, where 1 indicates True and -1 means false. Note that common input types may include binary, categorical, or numerical features, all of which are discretized and binarized to form our binary input features. The hidden layer has $K$ neurons, where $K$ determines the total number of rules and serves as a hyperparameter. The first layer is referred to as the Rule Layer, and the output layer as the OR layer.

**Rule Layer**: Each neuron in the hidden layer is denoted as $o_k \in \{0, 1\}$. The output represents whether a rule is satisfied. Each neuron is calculated as:

$$o_k = \mathbb{1}\left\{ \sum_{d=1}^{D} w_{d,k} x_d - \sum_{d=1}^{D} |w_{d,k}| = 0 \right\}, \quad \text{for } k = 1, \ldots, K \tag{1}$$

where $\mathbb{1}\{\cdot\}$ denotes the indicator function. Note that the indicator function is one if and only if all inputs match the sign of the corresponding weights: all positive weights should have the inputs of 1, and all negative weights should have the inputs of -1, and zero weights mean that the corresponding inputs are excluded from the rule.

Note that the indicator function is non-differentiable, which will make the gradient hard to compute. Here, we will instead use a differentiable smooth function to approximate the indicator function (this will also ease the theoretical analyses). For example, we can use $h(u) = \exp\left(-\frac{u^2}{\tau}\right)$, $\tau > 0$, to approximate $\mathbb{1}\{u = 0\}$, where $\tau$ is the tunable temperature and controls the approximation error.

**OR Layer**: The second layer operation is defined as

$$f(x) = \sum_{k=1}^{K} u_k o_k \tag{2}$$

where we assume that the rules contribute to the final prediction in a weighted additive form to reflect the OR composition, where $u_k$ denotes the weight assigned to the $k$-th rule.

### 3.2 Hypernetwork: Generate Network Weights

Define the network weights for the previous model as $\theta = (w, u)$, where $w \in R^{D \times K}$, and $u \in R^K$. Instead of learning only one set of $\theta$, we will learn the distribution of $\theta$, denoted as $\mu$. Specifically, we introduce a generative model as the hypernetwork to produce samples of $\theta$, i.e.,

$$\theta = G(\epsilon), \quad \text{where } \epsilon \sim p(\epsilon), \quad \theta \in R^{(D+1)K}. \tag{3}$$

Here, $\epsilon$ is drawn from some simple base distribution such as Gaussian distribution. More details about the hypernetwork structure can be seen in Appendix C. Note that, in this way, our model parameters have been changed to $\mu$, which is defined as the distribution of $\theta$, and the proposed Hyperlogic model is

$$f_\mu(x) = \mathbb{E}_\mu \left[ \sum_{k=1}^{K} u_k h \left( \sum_{d=1}^{D} w_{d,k} x_d - \sum_{d=1}^{D} |w_{d,k}| \right) \right]. \tag{4}$$

In practice, we can use the Monte Carlo method to estimate the above expectation by randomly drawing $M$ samples from $\mu$, denoted as $\theta^1, \ldots, \theta^M$, and we define

$$f_M(x) = \frac{1}{M} \sum_{m=1}^{M} \left[ \sum_{k=1}^{K} u_k^m h \left( \sum_{d=1}^{D} w_{d,k}^m x_d - \sum_{d=1}^{D} |w_{d,k}^m| \right) \right]. \tag{5}$$

This model can be regarded as a mixture of expert models with finite $M$ experts. Each expert is a shallow two-layer neural network that encodes at most $K$ if-then rules.

### 3.3 Expand to High-Dimensional Data

When working with high-dimensional data (e.g., 5000 dimensions), the number of parameters in the main network increases proportionally with the input dimensions, even if the network structure remains unchanged. This makes training a hypernetwork to generate weights for each module of the main network more challenging. To tackle this issue, we considered two strategies:

**Combining Original Weights with Hypernetwork-Generated Weights**    Instead of allowing the hypernetwork-generated weights ($W_{\text{hyper}}$) to fully dictate the main network's weights, we combine them with the original weights ($W_{\text{main}}$). The final weights are calculated as follows:

$$W_{\text{main-final}} = \alpha \cdot W_{\text{hyper}} + (1 - \alpha) \cdot W_{\text{main}},$$

where $\alpha$ is a learnable parameter constrained between 0 and 1. This strategy enhances stability without sacrificing the hypernetwork's ability to produce diverse weights. If the hypernetwork is poorly trained and generates inappropriate weights, the model adjusts $\alpha$ toward 0, effectively reverting to the standard main network without hypernetwork influence. We employed this strategy in our subsequent experiments.

**Generating Weights for Only Some Modules of the Main Network**    We test this strategy when the dataset dimension and size are very large in the supplementary experiment. It shows that generating only part of the main network's weights using the hypernetwork still significantly improves results. For more details, please see the supplementary experiments in Appendix B.

### 3.4   Loss Function

One can learn $\mu$ by minimizing the loss function $\ell(y, f_\mu(x))$, where $f_\mu(x)$ can be approximated by the finite-sample estimator in Eq. (5). Our loss contains two parts. One is the task-related loss function, denoted as $\ell_{task}(y, f_\mu(x))$. Suppose the problem is a binary classification problem, one can use the binary cross-entropy loss (BCE), where we first map $f_\mu(x)$ to a probability value between 0 and 1 using a link function such as sigmoid and the loss is defined as the negative log-likelihood of a sample belong to a class. Here, the task-related loss measures the prediction accuracy.

The second part of the loss is the regularization loss, denoted as $\ell_{reg}$, which is introduced to encourage the diversity of the generated $\theta$ and model sparsity (rule simplicity for each expert), i.e.,

$$\ell_{reg} = \lambda_1 D_{\text{KL}}(\mu \| \mu_0) + \lambda_2 \mathbb{E}_\mu[|u|]. \tag{6}$$

For the first regularization term, we introduce a prior distribution $\mu_0$ with a high entropy, and we minimize the relative entropy or KL divergence of the $\mu$ and $\mu_0$. Minimizing the relative entropy encourages the diversity of the generated $\theta$. For the second regularization term, we are considering the $\ell_1$ sparsity norm for the second OR layer's weights, where we hope that for each expert, the discovered number of rules is as compact as possible. In our paper, $\ell_{reg}$ represents the loss incurred by the hypernetwork, as illustrated in Fig. 1.

## 4   Theoretical Analysis

We will employ theoretical analysis to demonstrate that while the main network (i.e., a two-layer shallow neural network) has low capacity, incorporating the hypernet idea significantly enhances the model's expressive power. Specifically, we will establish that the proposed HyperLogic model serves as a universal approximator in the Barron space [19], which will be defined later. For simplicity, but without loss of generality, we consider the case where the hypernetwork fully generates the weights and the main network itself has no parameters. Additionally, we will present the generalization error under the above two types of regularization.

**Reparametrization:** To ease the proof, let's first rewrite Eq. (5) as

$$f_M(x) = \frac{1}{M} \sum_{m=1}^{M} \left[ \sum_{k=1}^{K} u_k^m h \left( w_k^{m\top} x \right) \right] \tag{7}$$

by reparametrization. To achieve this, we first use the split variable trick by defining $w^+ = \max\{w, 0\}$ and $w^- = -\min\{w, 0\}$, and reparametrize $w = w^+ - w^-$ and $|w| = w^+ + w^-$. In this way, we can get rid of the absolute value and have

$$\sum_{d=1}^{D} w_{d,k} x_d - \sum_{d=1}^{D} |w_{d,k}| = \sum_{d=1}^{D} w_{d,k}^+ (x_d - 1) + \sum_{d=1}^{D} w_{d,k}^- (-x_d - 1). \tag{8}$$

Therefore, we can simply reparametrize $w_k$, where $w_k \geq 0$, as a concatenation of the vector $[w_{d,k}^+]_{d=1,...,D}$ and $[w_{d,k}^-]_{d=1,...,D}$. We construct the new input data $x$ by making a copy of the original data and modifying each copy given Eq. (8). That is, for one copy, we subtract it by 1, and for another copy, we flip the sign and subtract it by 1. Then, we concatenate the two copies of data.

Given such a reparametrization, we get a simple form as Eq. (7). In the following analysis, we will still assume that $w_k \in R^D$ and $x \in R^D$, which doesn't affect the generality.

**Preparation for the Theoretical Analysis:** Our analysis below is based on the following observations. Given the generated random variable $((w_1, u_1), \ldots, (w_K, u_K)) \in \mathbb{R}^{(D+1)K}$, we have denoted their joint distribution as $\mu$. Let us further denote the marginal distribution of $(w_k, u_k)$ as $\mu_k$, $k = 1, \ldots, K$, respectively. Define a random variable $(\tilde{w}, \tilde{u}) \in \mathbb{R}^{D+1}$, which draws a sample from $\mu_k$ with (equal) probability $1/K$, $k = 1, \ldots, K$, and then scale the $u$-component by $K$. Denote the resulting mixture distribution as $\tilde{\mu}$. Then it follows from the linearity of expectation that

$$f_\mu(x) = \mathbb{E}_{((w_1,u_1),\ldots,(w_K,u_K))\sim\mu} \left[ \sum_{k=1}^K u_k h\left(w_k^\top x\right) \right]$$

$$= \mathbb{E}_{((w_1,u_1),\ldots,(w_K,u_K))\sim\mu} \left[ \frac{1}{K} \sum_{k=1}^K \tilde{u}_k h\left(w_k^\top x\right) \right] = \mathbb{E}_{(\tilde{w},\tilde{u})\sim\tilde{\mu}} \left[ \tilde{u} h\left(\tilde{w}^\top x\right) \right]. \qquad (9)$$

We define $\tilde{f}_{\tilde{\mu}}(x) := \mathbb{E}_{(\tilde{w},\tilde{u})\sim\tilde{\mu}} \left[ \tilde{u} h\left(\tilde{w}^\top x\right) \right]$ and we have shown that $f_\mu(x) = \tilde{f}_{\tilde{\mu}}(x)$ as stated above.

The motivation for introducing $\tilde{f}_{\tilde{\mu}}(x)$ is to facilitate the analysis of approximation error and generalization error. There exist theoretical results for a two-layer neural network of the form $g_M(x) = \sum_{m=1}^M u^m h\left(w^{m\top} x\right)$, where $h(\cdot)$ belongs to certain classes of smooth activation functions, and $(w^m, u^m)$, $m = 1, \ldots, M$, are $M$ independent samples drawn from a fixed distribution. In contrast, our proposed HyperLogic model involves randomly drawing samples $((w_1, u_1), \ldots, (w_K, u_K))$ from $\mu$, each with $K$ components. To align this with the existing theoretical framework, we perform a transformation as shown in Eq. (9). This involves converting the process of drawing $M$ samples, each with $K$ components, into drawing $MK$ samples, each with a single component. This reformulation maintains the same expectation results, allowing us to leverage existing theoretical results to analyze HyperLogic. Note that the conversion mentioned is purely for theoretical proof purposes and is not implemented in the actual algorithm.

### 4.1 Approximation Error of Finite Experts

Using the connection as shown in Eq. (9), we can directly leverage the approximation error results for a single hidden layer neural network. Let's first define the Barron space, which provides a set of functions for which neural networks can achieve good approximation properties.

**Definition 1** (Barron Space [19]). A function $f : \mathbb{R}^D \to \mathbb{R}$ belongs to the Barron space $\mathcal{B}$ if it can be represented as:

$$f(x) = \int_{\mathbb{R}\times\mathbb{R}^D\times\mathbb{R}} uh\left(w^\top x + b\right) d\mu(u, w, b)$$

where $h$ is an activation function, $\mu$ is a probability measure on $\mathbb{R} \times \mathbb{R}^D \times \mathbb{R}$, and the following Barron norm is finite:

$$\|f\|_{\mathcal{B}} = \inf \left\{ \int_{\mathbb{R}\times\mathbb{R}^D\times\mathbb{R}} |u| \|(w, b)\| d\mu(u, w, b) : f(x) = \int_{\mathbb{R}\times\mathbb{R}^D\times\mathbb{R}} uh\left(w^\top x + b\right) d\mu(u, w, b) \right\}.$$

Note that the representation of $f$ in the form $f(x) = \int uh\left(w^\top x + b\right) d\mu(u, w, b)$ may not be unique. The Barron norm seeks the representation that minimizes the integral of $|u| \|(w, b)\|$. The introduction of the Barron norm provides a quantitative measure of the "complexity" of a function in the context of neural networks. Functions with a smaller Barron norm are considered simpler and are easier to approximate with neural networks. Next, let's provide the approximation error analysis for our HyperLogic model (the proof can be found in Appendix D):

**Theorem 1.** *For any function $f$ in the Barron space, there exist $M$ experts $((w_1^m, u_1^m)$ $,\ldots, (w_K^m, u_K^m)) \in \mathbb{R}^{(D+1)\times K}$, $m = 1, \ldots, M$, forming a predictor $f_M$ as shown in Eq. (7), such that*

$$\|f - f_M\|_{L^2} \leq \frac{\|f\|_{\mathcal{B}}}{\sqrt{MK}},$$

*where the $L^2$ norm for a function $f : \mathbb{R}^D \to \mathbb{R}$ is defined as $\|f\|_{L^2} = \left(\int_{\mathbb{R}^D} |f(x)|^2 d\mathcal{D}(x)\right)^{\frac{1}{2}}$ where $\mathcal{D}(x)$ is the probability distribution over the input data.*

The above theorem asserts that a HyperLogic model (essentially a mixture of expert models) can approximate any continuous function in the Barron space to arbitrary precision given enough $M$, $K$ and appropriate weights.

## 4.2 Generalization Error: Entropic Regularization and Sparse Regularization

Let's derive the generalization error bounds for the HyperLogic model under entropic regularization (the first term of Eq. (6)) and sparse regularization (the second term of Eq. (6)). The key steps involve calculating the Rademacher complexity of the model class and then using this to bound the generalization error. The proof can be found in Appendix E. Let's give the results here.

**Theorem 2.** *For the function class $\mathcal{F}_{\mathrm{KL}} := \{f_\mu(\cdot) : D_{\mathrm{KL}}(\mu \| \mu_0) \leq B_{\mathrm{KL}}\}$, we have*

$$\mathbb{E}_{\mathcal{D}}[\ell(f)] - \hat{\ell}(f) \leq 2\sqrt{\frac{2B_{\mathrm{KL}}}{n}} + C\sqrt{\frac{\log(1/\delta)}{2n}}. \tag{10}$$

*Similarly, for the function class $\mathcal{F}_1 := \left\{f_\mu(\cdot) : \mathbb{E}_\mu\left[\frac{1}{K}\sum_{k=1}^{K} |u_k|\right] \leq B_1\right\}$, we have*

$$\mathbb{E}_{\mathcal{D}}[\ell(f)] - \hat{\ell}(f) \leq 64 B_1 \sqrt{(D+1)/n} + C\sqrt{\frac{\log(1/\delta)}{n}}.$$

*Here, $\mathbb{E}_{\mathcal{D}}[\ell(f)]$ is the expected loss over data distribution, $\hat{\ell}(f)$ is the empirical loss, $n$ is the number of samples, $C$ is a constant dependent on the loss function, and $\delta$ is the confidence level.*

From the results, we see that the relative entropy regularization effectively balances fitting the training data and adhering to the prior distribution with high entropy. Increasing the sample size or decreasing the KL bound enhances the model's generalization ability, ensuring good performance on unseen data. Similar conclusions can be arrived for the sparse regularization.

## 5 Experiment

In this section, we report experimental results to answer the following questions:

- **RQ1:** How does the performance of the optimal rule set selected by HyperLogic compare to the rule sets obtained by other methods?
- **RQ2:** How rich are the rule sets generated by HyperLogic, and how are their accuracy and diversity affected by parameters?
- **RQ3:** Can we further leverage the advantages of HyperLogic through ensemble learning to enhance performance?

### 5.1 Experiment Setup

**Implementation Details:** During training, for each data batch, we randomly generate $M_1$ samples of network weights to approximate the expectation (as shown in Eq. (5), here we compute $f_{M_1}(x)$ as an approximation in the training stage). Increasing $M_1$ enhances the stability of hypernetwork training but raises computational costs. In our experiments, $M_1$ is set to 5 or 10 . After training, we generate $M_2$ sets of weights from the hypernetwork, resulting in $M_2$ rule sets. In other words, in the inference stage, we use $f_{M_2}(x)$ instead. While $M_1$ is relatively small, $M_2$ can be large (e.g., 5000). We then select the rule set with the highest training accuracy as the optimal set, though other criteria, such as minimal loss or custom evaluation metrics balancing accuracy and complexity, can also be used.

For HyperLogic, We use Adam as the optimizer, and the learning rate is $1 \times 10^{-4}$, with weight decay is $1 \times 10^{-4}$. The number of training epochs is 10000. In the experiments for selecting the optimal rules for comparison, we set hyperparameter $M_1 = 5$, $M_2 = 5000$, $\lambda_1 = 0.01$, and $\lambda_2 = 0.1$ ($\lambda_1$ and $\lambda_2$ are related to the hypernetwork loss or regularization loss, as defined in Eq. (6)). In the following experiments that analyze the influence of hyperparameter, we adjusted only the corresponding hyperparameter while keeping others unchanged.

**Datasets:** We selected four publicly available binary classification datasets: MAGIC gamma telescope (magic), adult census (adult), FICO HELOC (heloc), and home price prediction (house). In all datasets, preprocessing was performed to encode categorical and numerical attributes as binary variables, which can be found in [17]. We compared our model HyperLogic with other state-of-the-art rule learning methods: DR-Net [17], CG [7], BRS [20], and RIPPER [6]. Since CG, BRS, and RIPPER cannot learn negative conditions, we additionally appended negative conditions for these models.

### 5.2 Performance Comparison

In the performance comparison section, we sampled $M_2 = 5000$ times from HyperLogic, selecting the rule set that performed best on the training set as the optimal rule set, and compared it with other methods. Table 1 presents the comparison of the accuracy of the optimal rule sets selected by

our method and those generated by other methods. Our method further improves upon DR-Net and outperforms other methods across all four datasets.

It is important to note that the rule set that performs best on the training set does not strictly guarantee the best performance on the test set, as shown in Fig. 2. However, this selection method is sufficiently simple, and experiments have shown that it can achieve good performance.

Table 1: Test accuracy based on a nested 5-fold cross-validation (%, mean $\pm$ standard error). Results corresponding to methods marked with * are directly sourced from [17].

| Method | Dataset | | | |
|---|---|---|---|---|
| | magic | adult | house | heloc |
| HyperLogic | **84.90** $\pm$ 0.73 | **83.11** $\pm$ 0.55 | **85.22** $\pm$ 0.62 | **71.03** $\pm$ 1.07 |
| DR-Net | 83.69 $\pm$ 0.55 | 82.95 $\pm$ 0.45 | 85.05 $\pm$ 0.51 | 70.07 $\pm$ 0.83 |
| CG* | 83.68 $\pm$ 0.87 | 82.67 $\pm$ 0.48 | 83.90 $\pm$ 0.18 | 68.65 $\pm$ 3.48 |
| BRS* | 81.44 $\pm$ 0.61 | 79.35 $\pm$ 1.78 | 83.04 $\pm$ 0.11 | 69.42 $\pm$ 3.72 |
| RIPPER* | 82.22 $\pm$ 0.51 | 81.67 $\pm$ 1.05 | 82.47 $\pm$ 1.84 | 69.67 $\pm$ 2.09 |

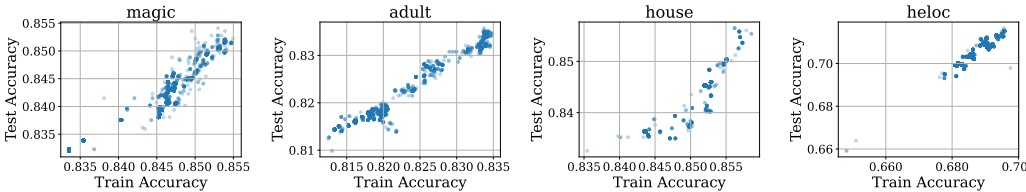

Figure 2: Train and test accuracies for sampled rule sets across four datasets

In addition to rule accuracy, we also considered two metrics to measure the compactness of the rules: model complexity and rule complexity. Model complexity is defined as the sum of the number of rules and the total number of conditions in the rule set; rule complexity is the average number of conditions in each rule of the model. Fig. 3 shows that our method reduces both model complexity and rule complexity compared to DR-Net, indicating that we have not only improved the accuracy of the rules but also made them more concise, demonstrating the effectiveness of our framework.

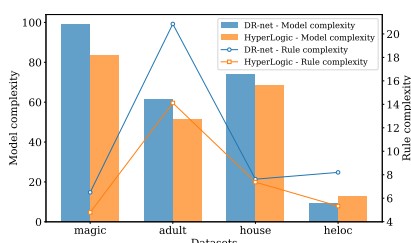

Figure 3: Model complexity and rule complexity comparison

### 5.3 Rule Analysis

In this section, we primarily considered the effects of three parameters, $M_1$, $\lambda_1$, and $M_2$, on the learned rules. Among these, $M_2$ primarily affects the selection of the final optimal rule set by controlling the number of candidate rule sets sampled from the hypernetwork after training phase, while $M_1$ and $\lambda_1$ directly influence the training of the hypernetwork.

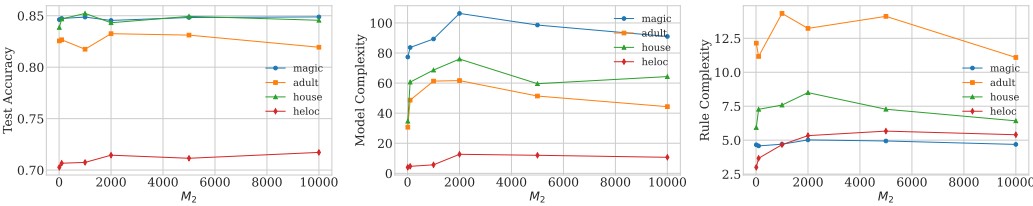

Figure 4: Analysis of the impact of $M_2$ on all datasets

For $M_2$, we considered the values 10, 100, 1000, 2000, 5000, and 10000 to examine its impact on the accuracy, model complexity, and rule complexity of the generated optimal rules, as shown in Fig.

4. It can be observed that the accuracy, model complexity, and rule complexity of the optimal rules generally increase with the increase of $M_2$, and then level off or slightly decrease. This is because, when $M_2$ is small, we may not be able to sample sufficiently suitable rule sets to fit the training set. When $M_2$ is too large, the sampled optimal rule set is more likely to overfit the training set, leading to a slight decrease in test accuracy.

So far, we have focused on the optimal rule set extracted from the hypernetwork. Table 2 further shows the different optimal results obtained in three training sessions for the heloc dataset, indicating that we can find different high-quality rule sets in different training sessions. However, considering only one optimal rule set is not enough to reflect the advantage of the hypernetwork in generating multiple high-quality rule sets in one training session, which provides more options and a deeper understanding of the data, something that methods learning only one set of rules cannot offer.

Table 2: Examples of optimal rule sets learned from different training runs on the heloc dataset

| Version | Rule | Train Acc | Test Acc |
|---|---|---|---|
| 1 | ①¬AvgFile $\leq$ 40.0 $\wedge$ ¬ExtRisk $\leq$ 69.0 $\wedge$ ¬PctNeverDelq $\leq$ 78.0 $\wedge$ NetFracBurden $\leq$ 77.0 | 69.28 | 71.51 |
| 2 | ①¬AvgFile $\leq$ 40.0 $\wedge$ ¬ExtRisk $\leq$ 66.0 $\wedge$ ¬OldTradeOpen $\leq$ 87.0 $\wedge$ ¬PctNeverDelq $\leq$ 78.0 $\wedge$ NetFracBurden $\leq$ 60.0 
 ②¬MaxDelq2Rec12M $\leq$ 5.0 $\wedge$ BankTradesHighUtil $\leq$ 2.0 $\wedge$ ¬ExtRisk $\leq$ 74.0 | 70.59 | 71.41 |
| 3 | ①¬ExtRisk $\leq$ 74.0 $\wedge$ ¬PctNeverDelq $\leq$ 92.0 $\wedge$ NetFracBurden $\leq$ 47.0 | 69.95 | 69.60 |

The ability of the hypernetwork to generate diverse rules is mainly related to $M_1$ and $\lambda_1$. $\lambda_1$, as the coefficient of diversity regularization term, directly affects the diversity of the hypernetwork output, while $M_1$ indirectly affects the process by influencing the stability of hypernetwork updates. For the impacts of $M_1$ and $\lambda_1$, we are no longer concerned with the optimal rule set but focus on all the rule sets generated by the hypernetwork, and describe the diversity and accuracy of the rule sets as a whole.

The diversity of the rules is measured in two aspects: 1. How many different rule sets can we generate by sampling a certain number of weights from the hypernetwork? 2. What is the degree of similarity between the generated different rule sets? We use Jaccard similarity to measure this.

We take the magic dataset as an example to analyze $M_1$ and $\lambda_1$; the results for the remaining datasets are in Appendix F. The impact of $M_1$ on rule diversity and test accuracy is shown in Fig. 5. It can be seen that when $M_1 = 5$, there is the best performance in all aspects. This may be because, when $M_1$ is small, the hypernetwork generates weights fewer times, making the training less stable. When $M_1$ is too large, the training and updates of the hypernetwork become too stable and conservative, slowing down the convergence speed and resulting in a performance decline of the hypernetwork.

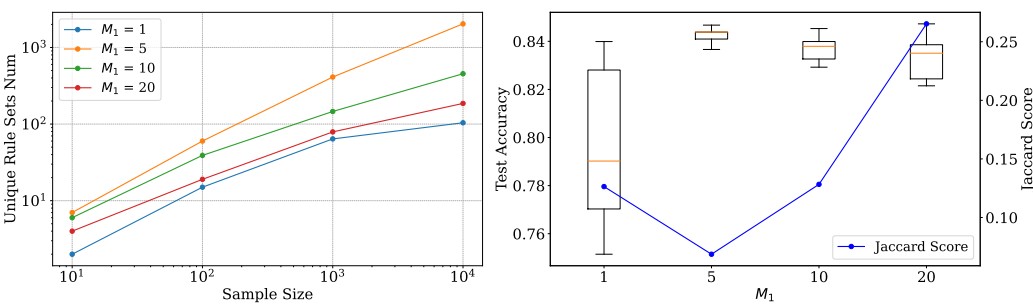

Figure 5: Analysis of the impact of $M_1$ on magic dataset

The impact of $\lambda_1$ on rule diversity and test accuracy is shown in Fig. 6. It can be seen that increasing $\lambda_1$ significantly enhances the diversity of the rule sets, but excessive emphasis on rule diversity may affect the accuracy of the rules.

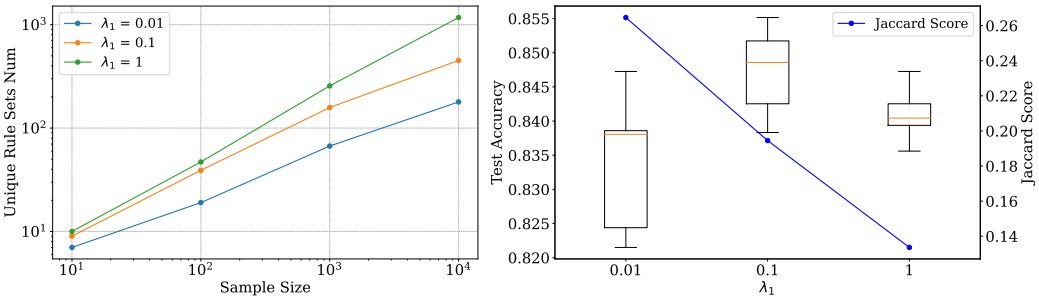

Figure 6: Analysis of the impact of $\lambda_1$ on magic dataset

## 5.4 Ensemble Learning

Since the hypernetwork can generate a diverse and rich set of rule sets, a natural idea is to use ensemble learning to achieve better classification performance. We use the simple averaging voting method for ensemble learning. We select the top $L$ rule sets based on their accuracy on the training set and report their accuracy on the test set. The values of $L$ are 1, 5, 10, 30, 50, 100, and 200. As shown in Fig. 7, the test accuracy initially increases with the increase of $L$, then slightly decreases.

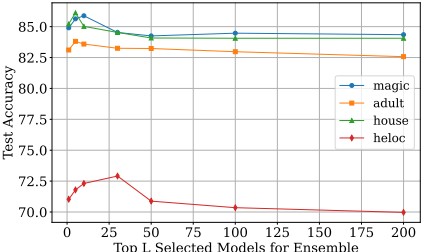

Figure 7: Test accuracy using top $L$ rule sets in ensemble learning.

This may because the top single rule set is already highly effective, leaving little room for improvement. When $L$ is too large, less effective rule sets decrease the ensemble's overall performance. Additionally, our current strategy focuses on balancing diversity and accuracy of single rule sets. Adjusting the strategy specifically for ensemble learning could yield better results.

## 6 Conclusion and Limitations

In this paper, we proposed HyperLogic, a novel framework that enhances the field of differentiable rule learning through the integration of hypernetworks. We also provided a theoretical foundation for HyperLogic's performance, explaining its effectiveness. This includes an analysis of approximation error and generalization capabilities under sparsity and diversity regularization. What's more, we conducted extensive experiments on multiple datasets, demonstrating that HyperLogic accelerates the training process while producing more concise and accurate rules.

Despite these advancements, HyperLogic introduces additional hyperparameters and requires further exploration of ensemble learning within this framework. Future research will focus on alternative training strategies, more stable weight combination methods, and applying HyperLogic to a broader range of datasets and tasks to enhance its generality and practicality.

## 7 Broader Impacts

The development and utilization of interpretable logic rules through HyperLogic have significant positive societal impacts, particularly in high-risk domains such as healthcare, finance, and legal systems. By generating multiple candidate rule sets, HyperLogic provides more flexible and comprehensive insights, enhancing transparency and trust in decision-making processes. This is crucial for ensuring safety and regulatory compliance. However, it is important to note that while our method provides richer rule sets and aids experts in understanding data patterns, these rules should complement rather than replace expert judgment. Relying solely on automated rules without expert oversight in critical areas could pose significant risks. Thus, our approach emphasizes the collaborative role of machine learning models and human experts to mitigate potential negative impacts.

## Acknowledgments

Shuang Li's research was in part supported by the National Science and Technology Major Project under grant No. 2022ZD0116004, the NSFC under grant No. 62206236, Shenzhen Stability Science Program 2023, Shenzhen Key Lab of Cross-Modal Cognitive Computing under grant No. ZDSYS20230626091302006, Longgang District Key Laboratory of Intelligent Digital Economy Security, and SRIBD Innovation Fund SIF20240010.

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

# A Compare with classical rule learning methods

We first compare HyperLogic with falling rule lists(FRL) methods like [39, 40, 41], which are often used in practice. FRL methods explicitly construct a falling list of rules with probabilities; Differentiable HyperLogic focuses on directly mining patterns and generating rules from data in a scalable manner using neural networks. Our differentiable approach demonstrates **better scalability for pattern mining tasks on large-scale data** (see Section 2). The main differences between FRL and the HyperLogic are:

1. **Rule Generation:**
   - FRL methods rely on other rule mining techniques to generate initial candidate rules.
   - HyperLogic uses a differentiable neural network approach to directly mine patterns and generate rules from data in an end-to-end manner.
2. **Scalability:**
   - FRL methods may face scalability issues when dealing with large candidate rule sets or complex data.
   - HyperLogic, a neural network-based approach, is more scalable for pattern mining in large-scale data.
3. **Rule Ordering:**
   - FRL methods explicitly order the rules into a descending list based on rule probabilities or other criteria.
   - HyperLogic generates an unordered set of rules.
4. **Probabilistic Interpretation:**
   - FRL methods provide probabilistic interpretations for the rules by construction.
   - HyperLogic does not directly output rule probabilities, although probabilities could potentially be derived from the learned patterns.
5. **Integration Potential:** A recent study [42] proposed a neural method to learn ordered rule lists, which could potentially be integrated with HyperLogic to enable joint rule generation and probabilistic ordering, combining the strengths of both approaches.

Other statistical rule mining methods like Bayesian rule lists [43] and Bayesian decision sets [20] provide probabilistic interpretations but are limited to binary classification and small rule sets. Fuzzy rule-based models [44] incorporate human-like reasoning but lack probabilistic predictions and scalability. In the supplementary experiments B, we include the current advanced traditional rule learning method CLASSY in the comparison to further demonstrate the advantages of our method over traditional methods.

# B Results for HyperLogic with DIFFNAPS

To expand our experiments to larger and more complicated cases, we considered the latest Neuro-Symbolic algorithm **DIFFNAPS** [30], which is capable of pattern mining under large-scale data conditions. For **HyperLogic**, we selected only the classifier part of DIFFNAPS as the main network to compare with vanilla DIFFNAPS.

## B.1 Large Synthetic Datasets

Following the original experiments, we tested the model's pattern mining performance under a **fixed input dimension of 5000** and varying total number of **categories K (ranging from 2 to 50)**, measured by the F1 score, with each category containing **1000 samples**.

Compared with the current data set, in our **new data set**, the feature dimension has been raised from a maximum of 154 dimensions to a maximum of **5k dimensions**, the amount of data has been raised from a maximum of 24,000 to a maximum of **50,000**, and the task has been raised from a maximum of 2 categories to a maximum of **50 categories**, reflecting the characteristics of the **task diversity and complexity**.

The experimental results are shown in the table below. It can be seen that in datasets with fewer categories, due to the smaller total number of samples, **HyperLogic** has not yet received sufficient training and does not perform ideally. However, in more challenging classification datasets with an increased number of samples, the model's performance has significantly improved, **with an average F1 score increase of 6%**. This fully demonstrates that our framework can empower diverse neural rule learning networks, capable of handling large-scale data and possessing a good range of applications.

| Dataset (K=) | DIFFNAPS | HyperLogic |
|:---:|:---:|:---:|
| 2 | **0.788 ± 0.080** | 0.726 ± 0.084 |
| 5 | **0.703 ± 0.030** | 0.675 ± 0.036 |
| 10 | 0.726 ± 0.013 | **0.751 ± 0.020** |
| 15 | 0.622 ± 0.017 | **0.800 ± 0.028** |
| 20 | 0.712 ± 0.015 | **0.760 ± 0.011** |
| 25 | 0.688 ± 0.020 | **0.770 ± 0.020** |
| 30 | 0.602 ± 0.014 | **0.766 ± 0.021** |
| 35 | 0.557 ± 0.014 | **0.668 ± 0.018** |
| 40 | 0.635 ± 0.022 | **0.712 ± 0.011** |
| 45 | 0.611 ± 0.009 | **0.694 ± 0.009** |
| 50 | 0.603 ± 0.010 | **0.702 ± 0.012** |

Table 3: The F1 score (± std) of two methods among 11 synthetic datasets

## B.2  Large Real Datasets

We evaluated our method on four large biological datasets following the settings of DIFFNAPS: Cardio, Disease, BRCA-N, and BRCA-S, using the area under the curve (AUC) as the metric. We continued to combine our approach (HyperLogic) with DIFFNAPS as the main network and compared it to vanilla DIFFNAPS. Additionally, FRL [39] cannot scale to non-trivial data, while CLASSY [41] was already compared in the original paper.

The table shows the dataset details (i.e. samples ($n$), features ($D$), and classes (K)), number of discovered patterns (#P), average pattern length (|P|), and AUC scores (results for DIFFNAPS and CLASSY are taken directly from [30]).

| Dataset | $n$ | $D$ | K | HyperLogic | | | DIFFNAPS | | | CLASSY | | |
|:---|:---:|:---:|:---:|:---:|:---:|:---:|:---:|:---:|:---:|:---:|:---:|:---:|
| | | | | #P | $\|P\|$ | AUC | #P | $\|P\|$ | AUC | #P | $\|P\|$ | AUC |
| Cardio | 68k | 45 | 2 | 15 | 2 | **0.57** | 14 | 2 | 0.56 | 10 | 2 | 0.36 |
| Disease | 5k | 131 | 41 | 866 | 2 | **0.86** | 838 | 2 | 0.84 | 25 | 2 | 0.11 |
| BRCA-N | 222 | 20k | 2 | 187 | 6 | **0.95** | 146 | 9 | 0.91 | 3 | 1 | 0.45 |
| BRCA-S | 187 | 20k | 4 | 1k | 2 | **0.89** | 1k | 2 | 0.86 | 2 | 1 | 0.23 |

Table 4: Comparison of HyperLogic, DIFFNAPS, and CLASSY across 4 real datasets.

CLASSY lacks the finesse to effectively mine patterns for large-scale real-world tasks. Moreover, despite competing with the strong DIFFNAPS baseline, HyperLogic achieved further improvements, demonstrating its potential for handling large, real-world datasets.

## C  Hypernetwork Details

We adopted HyperGAN as our hypernetwork. Assuming the main network comprises $N$ distinct weight partitions, HyperGAN includes a mixer $Q$ and $N$ generators $G_1, G_2, \ldots, G_N$.

**Mixer $Q$**: Receives high-dimensional Gaussian distributed random samples $s \sim \mathcal{N}(0, I)$, and transforms it into a $N \times h$ dimensional vector $z$, further split into $N$ samples of $h$-dimensional vectors $z_1, z_2, \ldots, z_N$. The mixer's design reflects the necessity for correlations between layer weights, as each layer's output becomes the subsequent layer's input.

**Generators $G_i$**: Each generator receives a vector $z_i$ from the mixer, producing an output vector of dimension $m_i$, where $m_i$ represents the parameter count for the $i$-th part of the network weights. These vectors are then reshaped to meet the specifications of the corresponding layers in the main network.

In our specific task, the number of generators $N$ is set to 2, and each generator's input dimension $h$ is 64. The input dimension of the Mixer, which is the sampled noise, is 256, and it produces an output of $N \times h = 128$. Each of the above models has two hidden layers with dimension 512, and the activation function is ReLu.

Our main network is designed to generate $K = 50$ rules. For a dataset with dimension $D$:

- **Generator 1:** Produces an output of dimension $(D \times K, 1)$, representing the Rule layer.
- **Generator 2:** Produces an output of dimension $(K \times 1, 1)$, representing the OR layer.

These outputs are then reshaped to meet the specifications of the corresponding layers in the main network.

# D  Proof for Theorem 1

*Proof.* Using Barron's theorem (e.g., [19, Theorem 11.3]), there exist weight coefficients $(w_k^m, \tilde{u}_k^m)$, $m = 1, \ldots, M$, $k = 1, \ldots, K$, such that the function $\tilde{f}$ defined by

$$\tilde{f}(x) = \frac{1}{MK} \sum_{m=1}^{M} \sum_{k=1}^{K} \tilde{u}_k^m h\left((w_k^m)^\top x\right)$$

satisfies

$$\|f - \tilde{f}\|_{L^2} \leq \frac{\|f\|_{\mathcal{B}}}{\sqrt{MK}}.$$

Setting $u_k^m = \tilde{u}_k^m / K$, $w^m = (w_1^m, \ldots, w_K^m)$, $u^m = (u_1^m, \ldots, u_K^m)$ yields the desired result. $\square$

# E  Proof for Theorem 2

## E.1  Generalization Error: Diverse Regularization

Let us compute the Rademacher complexity of the model class

$$\mathcal{F}_{\mathrm{KL}} := \left\{ f_\mu(\cdot) : D_{\mathrm{KL}}(\mu \| \mu_0) \leq B_{\mathrm{KL}} \right\},$$

where $\mu_0$ is a product distribution $\tilde{\mu}_0^{\otimes K}$. Using the property of relative entropy, for each pair of marginal distributions $\mu_k$ and $\tilde{\mu}_0$, $k = 1, \ldots, K$, their relative entropy satisfies $D_{\mathrm{KL}}(\mu_k \| \tilde{\mu}_0) \leq B_{\mathrm{KL}}$. Using the convexity of the relative entropy, the mixture distribution $\tilde{\mu}$ satisfies $D_{\mathrm{KL}}(\tilde{\mu} \| \tilde{\mu}_0) \leq B_{\mathrm{KL}}$. Hence, the function class $\mathcal{F}_{\mathrm{KL}}$ belongs to the function class

$$\tilde{\mathcal{F}}_{\mathrm{KL}} := \left\{ \tilde{f}_{\tilde{\mu}}(\cdot) : D_{\mathrm{KL}}(\tilde{\mu} \| \tilde{\mu}_0) \leq B_{\mathrm{KL}} \right\}.$$

Using [19, Corollary 10.17], the Rademacher complexity of $\tilde{\mathcal{F}}_{\mathrm{KL}}$ is bounded by $\sqrt{2B_{\mathrm{KL}}/n}$. Given our bound on the Rademacher complexity, for $f \in \tilde{\mathcal{F}}_{\mathrm{KL}}$, we get:

$$\mathbb{E}_{\mathcal{D}}[\ell(f)] - \hat{\ell}(f) \leq 2\sqrt{\frac{2B_{\mathrm{KL}}}{n}} + C\sqrt{\frac{\log(1/\delta)}{2n}}. \tag{11}$$

where $\mathbb{E}_{\mathcal{D}}[\ell(f)]$ is the expected loss, $\hat{\ell}(f)$ is the empirical loss, $n$ is the number of samples, $C$ is a constant dependent on the loss function, and $\delta$ is the confidence level.

## E.2  Generalization Error: Sparse Regularization

Next, we will derive the generalization error bounds for the HyperLogic model under sparse regularization (as shown in the second term of Eq. (6)).

Let us compute the Rademacher complexity of the model class

$$\mathcal{F}_1 := \left\{ \tilde{f}_\mu(\cdot) : \mathbb{E}_\mu\left[\frac{1}{K} \sum_{k=1}^{K} |u_k|\right] \leq B_1 \right\}.$$

Observe that the constraint $\mathbb{E}_\mu\left[\frac{1}{K} \sum_{k=1}^{K} |u_k|\right] \leq B_1$ implies that

$$\mathbb{E}_{\tilde{\mu}}[|\tilde{u}|] = \mathbb{E}_\mu\left[\left|\frac{1}{K} \sum_{k=1}^{K} u_k\right|\right] \leq \mathbb{E}_\mu\left[\frac{1}{K} \sum_{k=1}^{K} |u_k|\right] \leq B_1.$$

Hence, the function class $\mathcal{F}$ belongs to the function class

$$\tilde{\mathcal{F}}_1 := \left\{ \tilde{f}_{\tilde{\mu}}(\cdot) : \mathbb{E}_{\tilde{\mu}}[|\tilde{u}|] \leq B_1 \right\}.$$

Using [19, Proposition 11.23], the Rademacher complexity of $\tilde{\mathcal{F}}_1$ is bounded by $32B_1\sqrt{(D+1)/n}$. The generalization error bound for the HyperLogic model under sparse regularization, derived from the Rademacher complexity, can be expressed as follows:

$$\mathbb{E}_{\mathcal{D}}[\ell(f)] - \hat{\ell}(f) \leq 64B_1\sqrt{(D+1)/n} + C\sqrt{\frac{\log(1/\delta)}{n}}$$

The explaination of this error bound is similar with the previous one.

# F    Supplementary Result

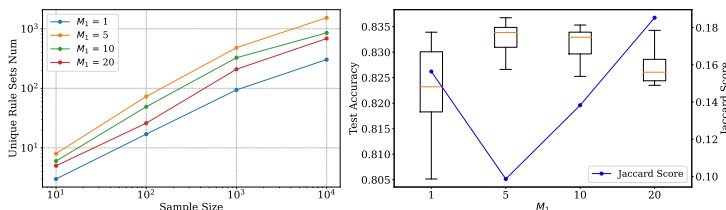

Figure 8: Analysis of the impact of $M_1$ on adult dataset

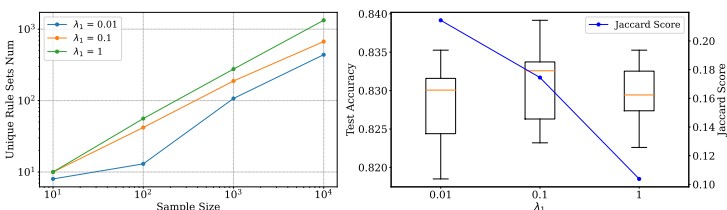

Figure 9: Analysis of the impact of $\lambda_1$ on adult dataset

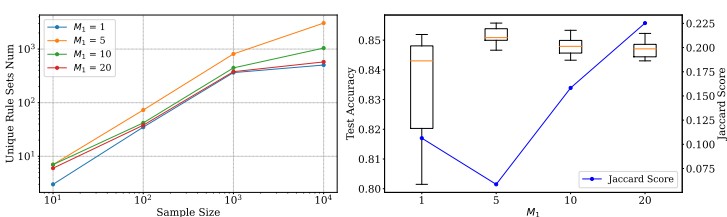

Figure 10: Analysis of the impact of $M_1$ on house dataset

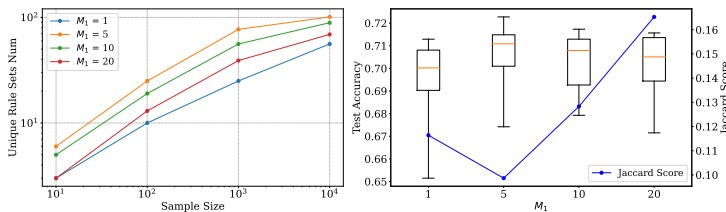

Figure 12: Analysis of the impact of $M_1$ on heloc dataset

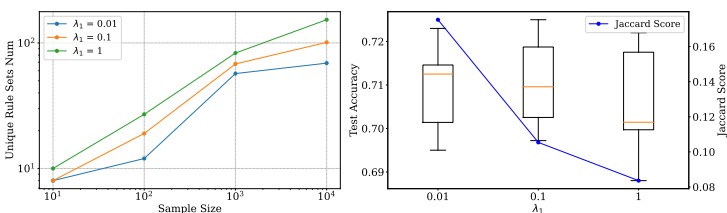

Figure 13: Analysis of the impact of $\lambda_1$ on heloc dataset

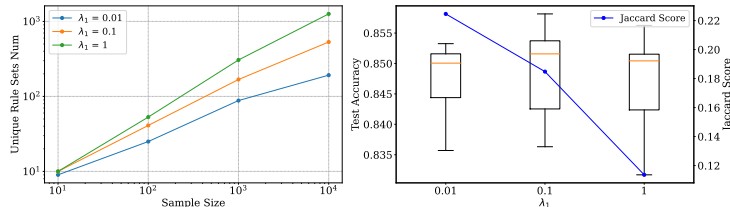

Figure 11: Analysis of the impact of $\lambda_1$ on house dataset

# G    Computing Infrastructure

For our method, all experiments were conducted on a Linux server with an Intel(R) Xeon(R) Gold 6248R CPU @ 3.00GHz and 30Gi of memory, running Ubuntu 20.04.5 LTS, using one of the NVIDIA GeForce RTX 3090 GPUs available on the server. Each experimental run took approximately 10-20 minutes to complete. This setup ensures that our experiments are reproducible and efficient.

