# OpenReview forum: "HyperLogic: Enhancing Diversity and Accuracy in Rule Learning with HyperNets"
_NeurIPS.cc/2024/Conference — NeurIPS 2024 poster_

### Official Review · Reviewer_hMMc · 2024-07-11

**Soundness:** 3
**Presentation:** 3
**Contribution:** 2
**Rating:** 5
**Confidence:** 3

**Summary:**

The paper presents HyperLogic, a pioneering framework that integrates hypernetworks into differentiable rule learning, enhancing the interpretability and flexibility of neural networks. It provides a strong theoretical foundation and backs it with extensive empirical evidence, demonstrating its effectiveness over traditional rule learning methods. HyperLogic's ability to generate diverse and accurate rule sets is particularly noteworthy, addressing a significant gap in the field of AI interpretability.

**Strengths:**

1. The paper introduces HyperLogic, a novel framework that integrates hypernetworks with rule-learning networks, which is a significant contribution to the field of interpretable machine learning.
2. The authors provide a solid theoretical analysis, including approximation error bounds and generalization capabilities, which substantiates the effectiveness of HyperLogic.
3. Comprehensive experiments on multiple datasets demonstrate the superiority of HyperLogic in learning diverse, accurate, and concise rules compared to existing methods.
4. The paper addresses a critical need for interpretability in high-stakes domains, offering a transparent decision-making process through the generation of logic rules.

**Weaknesses:**

1. The introduction of additional hyperparameters may lead to increased sensitivity, which could impact the model's performance and generalizability.
2. The framework's complexity might pose challenges for practitioners not well-versed in advanced neural network architectures.  While the paper mentions the efficiency of the rule-learning process, the computational overhead introduced by the hypernetworks is not thoroughly discussed.
3. The framework's applicability to a broader range of datasets and tasks needs to be investigated to ensure its generality and practical utility.

**Questions:**

How does HyperLogic perform in domains outside of the tested datasets, particularly in non-binary classification tasks?

**Limitations:**

Applications to more general scenarios.

---

> ### Author Rebuttal · Authors · 2024-08-07
>
> **Summary**
>
> We are grateful that reviewer hMMc has a positive impression of the high-level design of our framework and the promising results of comprehensive experiments. To address your questions about the details of our method, we provide point-wise responses as follows.
>
>
> **Q1. The introduction of additional hyperparameters may increase sensitivity, which could impact the model's performance and generalizability.**
>
> The additional hyperparameters brought by our framework mainly include the hypernet architecture, the coefficient lamba1 of the diversity loss caused by the hypernet in the loss function, and the number of weight sampling times $k_1$ and $k_2$ during and after training. Among them, adding more layers in the architecture adjustment and the number of sampling times $k_1$ after training have little impact on the training, while the coefficient $\lambda_1$ and the number of sampling times $k_1$ during training have a more significant impact on the training results, which requires us to have a good choice of hyperparameters. Further strategies. Furthermore, we additionally consider an adaptive weight fusion mechanism to optimize model performance through weighted fusion between the weights generated by the hypernet and the original main network weights. This mechanism is controlled by a learnable parameter α, which can be automatically adjusted during the training process to deal with the instability caused by improper selection of hyperparameters. (For more details, please refer to Q1 of Reviewer Q4Et.)
>
> **Q2. The framework's applicability to a broader range of datasets and tasks needs to be investigated.**
>
> We extended the experiment and proved that our HyperLogic framework can choose other differentiable rule learning algorithms besides DRNet as the main network combination to solve more tasks, such as processing high-dimensional large-scale data with multiple tasks. Classification tasks reflect the broad application scenarios of our framework.
>
> Specifically, the core feature of our framework, HyperLogic, is as an inclusive framework to allow enhancements to all neural methods as long as they are relevant to learning a set of interpretable weights end-to-end. This means that we can choose different differentiable rule learning networks as the main network according to needs to handle different tasks. Currently, for simplicity of presentation, our main network is derived from DRNet, which only supports binary classification tasks and is very time-consuming, preventing it from being proven on larger-scale data. To prove that our framework can handle larger data sets, we replace the rule learning model with the latest DIFFNAPS [1], which can perform multi-classification tasks in high-dimensional data.
>
> Compared with the current data set, in our **new data set**, the feature dimension has been raised from a maximum of 154 dimensions to a maximum of **5k dimensions**, the amount of data has been raised from a maximum of 24,000 to a maximum of 50,000, and the task has been raised from a maximum of 2 categories to a maximum of **50 categories**, reflecting the characteristics of the **task diversity and complexity**. In this case, HyperLogic-enhanced DIFFNAPS showed **a further average 6.1% F1 score improvement compared to the already good vanilla DIFFNAPS**, proving the effectiveness of the framework (please refer to General Q1 of global response for task details).
>
> [1] Walter, N. P., Fischer, J., & Vreeken, J. (2024, March). Finding interpretable class-specific patterns through efficient neural search. In Proceedings of the AAAI Conference on Artificial Intelligence (Vol. 38, No. 8, pp. 9062-9070).
>
> **Q3. The computational overhead introduced by the hypernetworks should be discussed more.**
>
> Since our framework introduces an additional super network, more computing load is inevitable. We take supplementary experiments as an example (please refer to General Q1 of global response for task details) to demonstrate the original algorithm DIFFNAPS and the main network based on DIFFNAPS The average F1 score, time and number of parameters of the HyperLogic framework in different tasks are shown in the table.
>
> | Method     | F1-score | Time(min) | #Params(W) |
> |------------|----------|-----------|------------|
> | DIFFNAPS   | 0.659    | 1.565     |1212.1      |
> | HyperLogic | 0.720    | 1.699     |5161.61     |
> |
>
> It can be seen that although the parameter increase is large, considering the significant improvement in F1 score and the extremely limited increase in time consumption, especially since Hyperlogic allows us to generate diverse rule sets in one training, the increase in computational load is extremely worthwhile.

---

> > ### Author Response · Authors · 2024-08-12
> >
> > Dear Reviewer hMMc,
> >
> > We are grateful for your time and effort in reviewing our paper and providing thoughtful feedback. We have carefully considered all of your comments and have responded to them accordingly.
> >
> > As we near the end of the author-reviewer discussion, we would like to hear your feedback on our rebuttal. We would greatly appreciate it if you could review our responses and let us know if we have adequately addressed your concerns. Additionally, we welcome any further comments or discussions you may have.
> >
> > Thank you for your valuable consideration.
> >
> > Best regards,
> >
> > The Authors

---

### Official Review · Reviewer_F7a8 · 2024-07-12

**Soundness:** 2
**Presentation:** 3
**Contribution:** 2
**Rating:** 6
**Confidence:** 5

**Summary:**

The authors consider the problem of learning interpretable rules for decision-making and propose a new neural rule learning framework to find these rules efficiently. In particular, they suggest to use a hypernetwork to learn a diverse set of network parameters for the rule-encoding network. On small binary data, they show that their algorithm finds reasonable rule sets that predict a given class well.

[Edit after rebuttal]: The authors provided additional discussion and experiments addressing my concerns, which leaves me with a *weak accept*.

**Strengths:**

- The presented approach seems novel and the considered problem is relevant.
- The presentation itself is clear and understandable.

**Weaknesses:**

- Despite the broad motivation of making neural decisions interpretable, the presented approach is very limited in what networks are used (2-layer specific architecture), which data is used (only binary datasets with labels), and which rules can actually be expressed (only conjunctions in antecedent, single label in consequent).
Especially the type of considered data needs to be communicated clearly from the start.
- There is a significant amount of related work, including recent neural approaches, that is not considered here. Some more classical approaches include statistical pattern mining [1] and falling rule lists [2,3,4]. Recently, a neural approach that finds differential patterns between labels -- which are essentially if-then rules has -- been proposed [5]. The experiments should be extended to properly compare to existing work.
- The experiments are very limited in terms of considered datasets: they are old “toy” datasets of small scale in both sample and feature dimension. Recent works, including [4], incorporate large-scale real-world data to actually show that such neural approaches are more scalable and can hence consider other data than classical approaches.

In summary, it is unclear to me whether the presented approach provides any improvement over existing work, or is scalable to large data at all.

[1] https://dl.acm.org/doi/10.1145/2783258.2783363
[2] https://proceedings.mlr.press/v38/wang15a.html
[3] http://proceedings.mlr.press/v84/chen18a/chen18a.pdf
[4] https://www.sciencedirect.com/science/article/pii/S0020025519310138
[5] https://ojs.aaai.org/index.php/AAAI/article/view/28756

**Questions:**

- In section 4, you convert the process of drawing M samples with K components each into drawing MK single-component samples. Does the latter actually match the former? It seems that the K components in the former are actually dependent, which would mean it does not correctly model it. Please clarify.
- In Table 1, the bold numbers are confusing. For adult, house, and heloc data, DR-Net, which is closely related and forms the basis of the architecture design, performs on par with the presented method. The results are within the given standard error.

**Limitations:**

While there is a brief discussion on limitations, neither the limitations in terms of data or rule language/architecture, nor the connection to existing work is properly discussed.

---

> ### Author Rebuttal · Authors · 2024-08-07
>
> **Summary:** We first thank reviewer F7a8 for the insightful comments, especially for the questions about our model design and experiment details, which helped us to clarify our paper. We would like to address the concerns one by one.
>
> **Q1. The method is limited in network structure (2-layer architecture), data format(only binary datasets with labels), and rule language(only conjunctions in antecedent, single label in consequent).**
>
> Our method is a general framework that can enhance many existing neural rule-learning methods, since the idea of ​​hypernet can be applied to different types of primary network, as long as they are related to learning a set of interpretable weights in a differentiable way (for some minor restrictions on the primary network architecture caused by the introduction of hypernet, please see Q6). The HyperLogic framework has no additional restrictions on data format and rule language, and only inherits from the selected primary network.
>
> As you mentioned, our implementation is currently limited in network structure, data format and rule language. But this is due to the limitations of DRNet, the primary network we refer to, not the limitations of our framework. We choose DRNet because the simplicity of the architecture is conducive to our proof and expression. However, considering the limitations of DR-Net, we can also consider other more flexible neural approach as the primary network to alleviate the restrictions on data format, rule language, etc. (please see Q2).
>
> **Q2. More related work, including recent neural approaches, should be considered.**
>
> Our proposed method introduces hypernetworks to generate weights for the main network, offering a novel framework. The main network is now a two-layer neural network specifically designed to retrieve rules directly from learned weights. However, other pre-existing neural methods can also be substituted, as long as they are related to end-to-end learning of a set of interpretable weights. Unlike traditional methods and recent Bayesian approaches, our method requires fewer hyperparameters and offers greater scalability. Differing from other neural methods, HyperLogic serves as a more versatile framework that can organically utilize these methods as the main network to enhance performance. Furthermore, our approach is capable of generating multiple sets of rule sets in a single training session, which is not feasible with other methods.
>
> Theoretically, the neural methods described in the previous section can serve as the main network for our framework, just like DRNet. Since DIFFNAPS is a novel algorithm capable of differentiable pattern discovery on large-scale data, **we have chosen to use DIFFNAPS as the main network in our supplemental experiments**. This expansion of our experiments further demonstrates the efficacy of our framework (refer to Q3 for details).
>
> **Q3. The experiments should be extended to properly compare to existing work.**
>
> Please refer to General Q1 in the global response.
>
> **Q4. In section 4, you convert the process of drawing M samples with K components each into drawing MK single-component samples. Does the latter actually match the former?**
>
> 1. The conversion mentioned is purely for theoretical proof purposes and is not implemented in the actual algorithm.
> 2. The preparation for the theoretical analysis (lines 175-191) remains valid regardless of the (in)dependence among the K components. Equation (9) is derived from the sample average of a distribution with equally weighted marginals.
> 3. To prove Theorem 1, we need to identify a set of parameters for equation (7) that satisfies the inequality stated in the theorem. One approach to achieve this is by using sampling weights transformed from Barron's theorem. It's important to note that this is just one possible set of parameters, and the actual learning algorithm does not utilize such sampling.
>
> **Q5. In Table 1, the bold numbers are confusing, as DR-Net performs similarly to the presented method on partial datasets within the standard error.**
>
> Our framework not only focuses on the accuracy of the optimal rule set, but also on the simplicity of the rule set and the richness of the learned candidate rule set.  Now we simply selected the best rule set on the training set for testing, and have surpassed the results of DRNet in a more concise rule set (Figure 3), demonstrating the effectiveness of our framework. At the same time, the optimal rule set on the training set is not completely equal to the optimal rule set on the test set (Figure 2), indicating that our training strategy (such as hyperparameter selection) and the final rule set selection strategy (such as further considering the complexity of the rule set) can further obtain higher test accuracy. We leave this for future work.
>
> **Q6. The limitations of network structure.**
>
> HyperLogic has almost no special restrictions on the architecture of differentiable rule learning networks, except for two possible problems, but both can be solved:
> 1. If the main network has too many parameters, the hypernetwork might struggle to produce stable weights. To solve this, we tried two approaches: ① Combine the original weights with those generated by the hypernetwork to improve stability while keeping the hypernetwork's flexibility (please refer to Q1 of Reviewer Q4Et for details). ② Only generate some of the main network's weights using the hypernetwork. Our tests show this still significantly improves results, even with large datasets (please see Q3 for details).
> 2. Some neural methods use a non-differentiable technique called weight pruning to simplify the model. To address this, we've chosen to keep all weights and apply a sigmoid function to them after processing. This makes the entire process differentiable. While this change might affect how the data is distributed and require adjustments to the model's settings, our tests show it doesn't harm performance.

---

> > ### Comment · Reviewer_F7a8 · 2024-08-08
> > **Rebuttal comment**
> >
> > Thank you for your response.
> >
> > Regarding *W1*: I appreciate the answer, the modularity of the approach did not become that evident in the original draft. I would suggest to rewrite the paper accordingly also incorporating the new results, as this strengthens the contribution.
> >
> > Regarding *W2*: The new results including their critical discussion does cover more recent neural approaches. However approaches such as the popular falling rule lists, which are heavily used in practice, should be compared to.
> >
> > Regarding *W3*: Still, not considering recent real datasets is a major weakness in the paper. The additional results only cover a larger synthetic dataset.
> >
> > I would like to encourage the authors to use the remaining time to (1) compare to falling rule lists and (2) consider at minimum one large real dataset, the concurrent literature on pattern mining does give plenty of examples of those.

---

> > > ### Author Response · Authors · 2024-08-12
> > > **Answers to new comments from Reviewer F7a8**
> > >
> > > Thank you for acknowledging our method and suggestions. We will clarify our approach's generality and contributions in the final draft. Additionally, we will compare our approach to falling rule lists and expand experiments to four large biological datasets.
> > >
> > > **1. Compare to falling rule lists (FRL) [2] and other methods**
> > >
> > > FRL methods explicitly construct a falling list of rules with probabilities; Differentiable HyperLogic focuses on directly mining patterns and generating rules from data in a scalable manner using neural networks. Our differentiable approach demonstrates **better scalability for pattern mining tasks on large-scale data** (see Section 2). The main differences between FRL and the HyperLogic are:
> > > 1. Rule Generation:
> > > - FRL methods rely on other rule mining techniques to generate initial candidate rules.
> > > - HyperLogic uses a differentiable neural network approach to directly mine patterns and generate rules from data in an end-to-end manner.
> > > 2. Scalability:
> > > - FRL methods may face scalability issues when dealing with large candidate rule sets or complex data.
> > > - HyperLogic, a neural network-based approach, is more scalable for pattern mining in large-scale data.
> > > 3. Rule Ordering:
> > > - FRL methods explicitly order the rules into a descending list based on rule probabilities or other criteria.
> > > - HyperLogic generates an unordered set of rules.
> > > 4. Probabilistic Interpretation:
> > > - FRL methods provide probabilistic interpretations for the rules by construction.
> > > - HyperLogic does not directly output rule probabilities, although probabilities could potentially be derived from the learned patterns.
> > > 5. Integration Potential:
> > > A recent study [1] proposed a neural method to learn ordered rule lists, which could potentially be integrated with HyperLogic to enable joint rule generation and probabilistic ordering, combining the strengths of both approaches.
> > >
> > > Other statistical rule mining methods like Bayesian rule lists [4] and Bayesian decision sets [5] provide probabilistic interpretations but are limited to binary classification and small rule sets. Fuzzy rule-based models [6] incorporate human-like reasoning but lack probabilistic predictions and scalability.
> > >
> > >
> > > **2. Consider large real datasets**
> > >
> > > We evaluated our method on four large biological datasets following the settings of DIFFNAPS[1]: Cardio, Disease, BRCA-N, and BRCA-S, using the area under the curve (AUC) as the metric. We continued to combine our approach (HyperLogic) with DIFFNAPS  as the main network and compared it to vanilla DIFFNAPS. Additionally, FRL[2] cannot scale to non-trivial data, while CLASSY was already compared in the original paper.
> > >
> > > The table shows the dataset details (i.e. samples (n), features (m), and classes (K) ), number of discovered patterns(#P), average pattern length(|P|), and AUC scores (results for DIFFNAPS and CLASSY are taken directly from [1]).
> > >
> > > ||||| HyperLogic | HyperLogic | HyperLogic | DIFFNAPS |DIFFNAPS| DIFFNAPS | CLASSY |CLASSY | CLASSY |
> > > |---------|-----|-----|----|:-----------:|------------|------|----------|-------|------|--------|-------|------|
> > > | Dataset | n   | m   | K  | #P| \|P\| | AUC  | #P| \|P\| | AUC | #P | \|P\| | AUC |
> > > | Cardio  | 68k | 45  | 2  | 15 | 2| 0.57 | 14 | 2| 0.56 |10| 2| 0.36 |
> > > | Disease | 5k  | 131 | 41 | 866 | 2 | 0.86 | 838 | 2| 0.84 | 25 | 2 | 0.11 |
> > > | BRCA-N  | 222 | 20k | 2  | 187 | 6  | 0.95 | 146 | 9| 0.91 | 3  | 1| 0.45 |
> > > | BRCA-S  | 187 | 20k | 4  | 1k | 2 | 0.89 | 1k | 2 | 0.86 | 2| 1| 0.23 |
> > >
> > > CLASSY lacks the finesse to effectively mine patterns for large-scale real-world tasks. Moreover, despite competing with the strong DIFFNAPS baseline, HyperLogic achieved further improvements, demonstrating its potential for handling large, real-world datasets.
> > >
> > > Please let us know if you have any other questions or suggestions for improving the final draft. We appreciate it a lot.
> > >
> > > [1] Walter, N. P., Fischer, J., & Vreeken, J. (2024, March). Finding interpretable class-specific patterns through efficient neural search. In Proceedings of the AAAI Conference on Artificial Intelligence (Vol. 38, No. 8, pp. 9062-9070).
> > >
> > > [2] Wang, F., & Rudin, C. (2015, February). Falling rule lists. In Artificial intelligence and statistics (pp. 1013-1022). PMLR.
> > >
> > > [3] Proença, H. M., & van Leeuwen, M. (2020). Interpretable multiclass classification by MDL-based rule lists. Information Sciences, 512, 1372-1393.
> > >
> > > [4] Yang, H., Rudin, C., & Seltzer, M. (2017, July). Scalable Bayesian rule lists. In International conference on machine learning (pp. 3921-3930). PMLR.
> > >
> > > [5] Wang, T., Rudin, C., Doshi-Velez, F., Liu, Y., Klampfl, E., & MacNeille, P. (2017). A bayesian framework for learning rule sets for interpretable classification. Journal of Machine Learning Research, 18(70), 1-37.
> > >
> > > [6] Jiménez, F., Sánchez, G., & Juárez, J. M. (2014). Multi-objective evolutionary algorithms for fuzzy classification in survival prediction. Artificial intelligence in medicine, 60(3), 197-219.A

---

> > > > ### Comment · Reviewer_F7a8 · 2024-08-12
> > > > **response to author's additional experiments**
> > > >
> > > > Thank you for these additional results addressing my concerns. This provides a much more comprehensive and convincing evaluation of your work. Even though HyperLogic only provides a marginal improvement over some existing methods in light of all experimental results, I do see its value for the community in a general framework as well as in a better theoretical analysis compared to what exists. Furthermore, the added results put it into perspective in concurrent work in the field and allow for a good and critical comparison.
> > > >
> > > > Consequently, I raise my score to *weak accept*, on the expectation that additional discussion on related work, results, and a critical analysis will be put into the revised manuscript.
> > > >
> > > > I wish the authors best of luck with the submission.

---

> > > > > ### Author Response · Authors · 2024-08-13
> > > > >
> > > > > Dear Reviewer F7a8,
> > > > >
> > > > > We would like to extend our gratitude once again for your insightful suggestions and positive response. We will also ensure the extra results and discussions will be updated in the revised manuscripts.
> > > > >
> > > > > Warm regards,
> > > > >
> > > > > The Authors

---

### Official Review · Reviewer_Q4Et · 2024-07-13

**Soundness:** 3
**Presentation:** 3
**Contribution:** 3
**Rating:** 6
**Confidence:** 2

**Summary:**

This paper introduces HyperLogic, a novel differentiable framework for rule learning using neural networks. Instead of directly training network weights, HyperLogic extracts rules by generating the weights for a primary network through hypernetworks. These hypernetworks create diverse sets of weights, functioning like a mixture of experts, thereby enhancing model flexibility. To train the hypernetworks, the authors minimize the relative entropy between the learned distribution and a prior distribution, incorporating regularization to encourage simplicity. They employ smooth approximations to ensure the problem be differentiable. The authors demonstrate HyperLogic’s effectiveness through both theoretical analysis and empirical results on multiple datasets.

**Strengths:**

- The paper presents an interesting rule-learning approach using hypernetworks, leveraging a set of experts to take advantage of complex and diverse models while maintaining interpretability.
- The authors provide a robust theoretical foundation for HyperLogic, showing the effectiveness of their approach through rigorous analysis.
- Extensive experiments on various datasets demonstrate that HyperLogic can learn multiple diverse and accurate rule sets, highlighting its performance.
- This paper is well-written and very easy to follow.

**Weaknesses:**

- The results seem to be sensitive to the choice of hyperparameters, which could affect the stability and generalizability of the method. Are there any guidelines or automated processes to aid in choosing the optimal hyperparameters?
- The scalability of the approach for extremely large datasets or high-dimensional data remains unclear. Further evaluation is needed to understand its performance in such scenarios.
- Typo: in line 125,  all negative weights should have the inputs of 0 ?

**Questions:**

see weaknesses

---

> ### Author Rebuttal · Authors · 2024-08-07
>
> **Summary**
>
> Many thanks to reviewer Q4Et for your positive comments and recognition of our contribution including an interesting framework, a robust theoretical foundation, comprehensive experiments, and a well-written paper. We would like to address your concerns one by one.
>
> **Q1. Are there any guidelines or automated processes to aid in choosing the optimal hyperparameters?**
>
> The additional hyperparameters brought by our framework mainly include the hypernet architecture, the coefficient lamba1 of the diversity loss caused by the hypernet in the loss function, and the number of weight sampling times $k_1$ and $k_2$ during and after training.
> 1) Regarding the hypernetwork architecture:
>
> We followed HyperGAN, which is a simple and effective hypernetwork architecture, and has been proven in experiments that more layers will not bring significant gains; explore other more powerful hypernetworks The architecture will be interesting as our future work.
>
> 2) Regarding the number of sampling times $k_2$ after training:
>
> $k_2$ only affects the results of our final extracted rules and does not directly affect the training of the model. Due to the simplicity constraint, the rule set we sampled is not easy to overfit. We can expand $k_2$ as much as possible to obtain a richer rule candidate set if resources and time allow.
>
> 3) Regarding the coefficient of diversity loss $\lambda_1$ and the number of sampling times $k_1$ in training:
>
> The two have a more significant impact on the effect. In practice, we can set a smaller initial value and gradually increase it until classification error staturates.
> Further, inspired by your question, we considered extending our model to alleviate the challenge of hyperparameter selection. We no longer let the weight $W_{hyper}$ generated by the super network completely act on the weight of the main network, but retain the weight $W_{main}$ of the main network, and combine it with the weight generated by the super network to obtain $W_{main-final} = \alpha* W_{hyper} + (1-\alpha)*W_{main}$, where $\alpha$ is a learnable parameter restricted to 0-1. This can increase the stability of our results without losing the advantage that the hypernetwork can generate diverse weights: in extreme cases, we set very inappropriate hyperparameters about the hypernetwork and generate inappropriate $W_{hyper}$, and the algorithm will Prompt alpha to approach 0, thereby offsetting the impact of inappropriate hyperparameter selection and allowing the model to degenerate into a normal version without a hypernet. The effectiveness of such improvements is demonstrated in supplementary experiments, especially for high-dimensional data. Updated algorithms and results will be added to the final version.
>
> **Q2. The scalability of the approach for extremely large datasets or high-dimensional data remains unclear.**
>
> Our framework aims to enhance various existing neural rule-learning methods, since hypernet ideas can be applied to different types of primary networks, as long as they are related to learning a set of interpretable weights in a differentiable manner. Therefore, HyperLogic is a general framework. However, it should be noted that the framework essentially stems from the main network, that is, the framework's task scope, model capabilities, etc. are still closely related to the main network, and scalability is often limited by the size of the primary network. However, we can easily switch to a more flexible primary network according to the needs of the task. This is just a simple adaptation, while the main ideas stay the same.
>
> Currently, for the sake of simplicity of description, our HyperLogic main network gets inspiration from DR-Net, which although has a simple architecture, only supports binary classification tasks and is very time-consuming, preventing HyperLogic from being proven on larger-scale data. To prove that our framework can handle larger data sets, we replaced HyperLogic's main network with the latest DIFFNAPS [1], which is a neural approach that can perform multi-classification tasks and pattern discovery in high-dimensional data.
>
> Under this new configuration, we conducted experiments on **a new dataset with feature dimensions up to 5,000, data size up to 50,000, and task types up to 50 categories**. In this case, HyperLogic, which uses DIFFNAPS as the main network, further demonstrated **an average 6.1% F1 score improvement** compared to the already good vanilla DIFFNAPS, while **the average time consumption only increased by 8.6%** (please refer to General Q1 in global response for task details), effectively proving the practical application potential of the HyperLogic framework in processing large-scale data sets and high-dimensional data.
>
> [1] Walter, N. P., Fischer, J., & Vreeken, J. (2024, March). Finding interpretable class-specific patterns through efficient neural search. In Proceedings of the AAAI Conference on Artificial Intelligence (Vol. 38, No. 8, pp. 9062-9070).
>
> **Q3. Typo: in line 125, do all negative weights have the inputs of 0?**
>
> Thank you for your feedback. We have reviewed line 125 and acknowledge that all negative weights should have the inputs of -1 . This error will be corrected in our revised manuscript.

---

> > ### Comment · Reviewer_Q4Et · 2024-08-12
> > **Official Comment by Reviewer Q4Et**
> >
> > Thanks for the detailed response. I have no further questions and would like to keep my positive score.

---

> > > ### Author Response · Authors · 2024-08-13
> > >
> > > Dear Reviewer Q4Et,
> > >
> > > Thank you for acknowledging our work. We will incorporate the relevant results in the final version to further enhance its quality.
> > >
> > > Best regards,
> > >
> > > The Authors

---

### Author Rebuttal · Authors · 2024-08-07

Dear esteemed reviewers,

We are grateful to all the reviewers for generously dedicating their time and effort to evaluating our paper. Their constructive feedback and valuable suggestions are helpful to further improve the quality of our work. We would like to answer general questions mentioned by all the reviewers here.

Thank you for your considerable contributions during the review process. We anticipate further discussions in the future.

Once again, many thanks to your valuable insights.

Best wishes,

The authors

**General Q1: The experiments should be extended to properly compare to existing work.**

To expand our experiments to larger and more complicated cases, we considered the latest Neuro-Symbolic algorithm **DIFFNAPS**, which is capable of pattern mining under large-scale data conditions. Following the original experiments, we tested the model’s pattern mining performance under **a fixed input dimension of 5000** and varying total number of **categories K (ranging from 2 to 50)**, measured by the F1 score, with each category containing **1000 samples**. For HyperLogic, we selected only the classifier part of DIFFNAPS as the main network to compare with vanilla DIFFNAPS.

Compared with the current data set, in our **new data set**, the feature dimension has been raised from a maximum of 154 dimensions to a maximum of **5k dimensions**, the amount of data has been raised from a maximum of 24,000 to a maximum of 50,000, and the task has been raised from a maximum of 2 categories to a maximum of **50 categories**, reflecting the characteristics of the **task diversity and complexity**.

The experimental results are shown in the table below. It can be seen that in datasets with fewer categories, due to the smaller total number of samples, Hyper Logic has not yet received sufficient training and does not perform ideally. However, in more challenging classification datasets with an increased number of samples, the model's performance has significantly improved, **with an average F1 score increase of 6%**. This fully demonstrates that our framework can empower diverse neural rule learning networks, capable of handling large-scale data and possessing a good range of applications.

***Table. The F1 score（+/- std） of two methods among 11 datasets***
| Dataset（K=） | DIFFNAPS         | HyperLogic       |
|---------------|------------------|------------------|
| 2             | 0.788  +/- 0.080 | 0.726 +/- 0.084  |
| 5             | 0.703  +/- 0.030 | 0.675 +/- 0.036  |
| 10            | 0.726  +/- 0.013 | 0.751 +/- 0.020  |
| 15            | 0.622 +/- 0.017  | 0.800 +/- 0.028  |
| 20            | 0.712 +/- 0.015  | 0.760 +/- 0.011  |
| 25            | 0.688 +/- 0.020  | 0.770 +/- 0.020  |
| 30            | 0.602 +/- 0.024  | 0.666 +/- 0.021  |
| 35            | 0.557 +/-  0.014 | 0.668 +/- 0.018  |
| 40            | 0.635 +/- 0.022  | 0.712  +/- 0.011 |
| 45            | 0.611 +/- 0.009  | 0.694 +/- 0.009  |
| 50            | 0.603 +/- 0.010  | 0.702 +/- 0.012  |

---

### Decision · Program_Chairs · 2024-09-25

**Decision:**

Accept (poster)

**Comment:**

The authors propose a novel approach for generating candidate rulesets using hyper-networks based on HyperGAN. The authors provided additional experimental results during the rebuttal, which, together with the results in the original submission, show competitive performance, and the ability to adopt their framework to different main models.